# A Survey of Dentists’ Perception of Chair-Side CAD/CAM Technology

**DOI:** 10.3390/healthcare9010068

**Published:** 2021-01-13

**Authors:** Mohammad Zakaria Nassani, Shukran Ibraheem, Enass Shamsy, Mahmoud Darwish, Asmaa Faden, Omar Kujan

**Affiliations:** 1Department of Restorative and Prosthetic Dental Sciences, College of Dentistry, Dar Al Uloom University, Riyadh 11512, Saudi Arabia; mznassani@dau.edu.sa; 2Department of Prosthetic Dental Sciences, AlFarabi College of Dentistry and Nursing, Riyadh 11691, Saudi Arabia; shukran.ibrahim@3alemni.eu (S.I.); dr_mdarwish@alfarabi.edu.sa (M.D.); 3Dental Health Care Department, Inaya Medical College, Riyadh 11352, Saudi Arabia; eshamssy@inaya.edu.sa; 4Department of Oral Medicine and Diagnostic Science, College of Dentistry, King Saud University, Riyadh 4545, Saudi Arabia; afaden@ksu.edu.sa; 5Oral Diagnostic and Surgical Sciences, UWA Dental School, University of Western Australia, Nedlands 6009, Australia

**Keywords:** chair-side, CAD/CAM, dentists, perception, Saudi Arabia, survey

## Abstract

The application of computer-aided design and computer-aided manufacturing (CAD/CAM) technology in dentistry has rapidly expanded. This survey aimed to investigate attitudes and current practice of dentists in Riyadh, Saudi Arabia regarding chair-side CAD/CAM technology. An online questionnaire was prepared and sent to a convenience sample of dentists in Riyadh. Questions related to practice aspects of chair-side CAD/CAM system, attitudes and training were presented. A total of 114 questionnaires were completed (77.6% response rate). Study population comprised dentists of a wide range of clinical experience who are working in the various regions of Riyadh city, Saudi Arabia. Of the participants, 27.2% indicated the presence of a chair-side CAD/CAM system at their current workplace and 57% used the chair-side CAD/CAM in the fabrication of dental crowns. The vast majority of participants (81%) believe that the overall quality of chair-side CAD/CAM restorations is at least as good as those fabricated by a lab technician or much better. Most responding dentists considered the chair-side CAD/CAM system as important in terms of time saving, boosting the number of visiting patients and income improvement. The willingness to dedicate the time and effort to learn the chair-side CAD/CAM technology was apparent amid a high proportion of dentists (75.4%). The findings of this pilot survey reflect broad satisfaction and positive attitude among the surveyed dentists towards use and outcome of chair-side CAD/CAM technology in the dental clinical practice. It seems that the CAD/CAM technology has infiltrated into the workflow of Saudi dental practices with speculations of growing implementation among the wider sector of dental practitioners in the future.

## 1. Introduction

Over the last two decades, the use and application of computer-aided design and computer-aided manufacturing (CAD/CAM) technology in dentistry has rapidly expanded [1]. The higher demand for rapid but effective esthetic dental treatment and metal-free restorations have led dentists to adopt dental procedures that save time and produce reliable and pleasing esthetic and functional results. In the 1960s, the computer-aided design and manufacturing was introduced for the use in aircraft and automotive industries [2]. Thereafter, in 1983, the CAD-CAM system continued to evolve and revolutionized restorative dentistry by the introduction of Chairside Economical Restoration of Esthetic Ceramics (CEREC) system (Sirona Dental Systems GmbH, Bensheim, Germany) [3]. The CEREC system is, historically, the first chair-side CAD/CAM system in the dental world [1]. This system has successfully connected a milling machine to a digital oral scanner and allowed production of dental restorations in a single appointment. In 2008, the E4D Dentist™ system was introduced and, similar to CEREC system, provided in-office dental restorations in a single visit [4]. According to methods of production, the CAD/CAM systems can be classified into three categories [5]: the chair-side system, the laboratory system and the centralized production. With the laboratory system and centralized production system, the responsibility of production of the dental restoration is delegated to the dental technician by the aid of CAD/CAM milling unit. A minimum of two visits to the dental office is required to complete the dental restoration using the former CAD/CAM systems. On the other side, the chair-side system allows the dentist to control the whole process beginning from taking a digital impression of the prepared tooth/teeth and then designing and production of the dental restoration/s. Lastly the final restoration is delivered at the same visit. Generally, CAD/CAM systems involve three elements [3]. The first element is a digitalization instrument/scanner that transforms geometry to digital data that can be processed by the computer. The second element is Software that processes the data obtained from the digital scanner. The third part of the CAD/CAM system is a milling machine that receives the information from the Software to produce a dental restoration with specific characteristics and design. To date, the CAD/CAM technology has been implemented to produce various types of dental restorations including inlays, onlays, crowns, veneers, multi-unit fixed partial dentures (FPDs), and implant abutments [1,2]. Moreover, the CAD/CAM technology has expanded its dental applications to comprise orthodontic treatment, fabrication of occlusal splints, fabrication of removable dentures and maxillofacial prostheses, guided implant surgery, orthognathic surgery, and guided bone regeneration [6,7,8,9,10,11,12,13,14,15]. This innovative technology is still evolving with growing and promising applications in the dental field. Research findings show that CAD/CAM fabricated dental restorations are of high-quality and may exceed the conventionally fabricated dental restorations in terms of accuracy and physical and mechanical properties [1,16,17,18,19,20]. Besides these merits, the CAD/CAM technology, particularly the chair-side system, offers dentists a number of advantages such as less dependence on the dental technician, less number of visits for the patient, simplified technical procedures, reduced consumption of materials, increased productivity, and cost-effective dental restorations [2]. However, the CAD/CAM technology has some limitations including high initial cost of purchasing the CAD/CAM system, time and cost investment to master the technique on the dentist/technician side, some difficulties in acquisition of accurate digital impressions for multiple units prosthesis., and a chance of fabrication errors or faulty shaping, especially with multi-unit dental restorations, which may risk the mechanical properties of the produced restoration/s [21,22].

So far the CAD/CAM technology has become an essential part of modern dentistry [23]. It can be speculated that this technology will change the shape of future dental practice. However, research to investigate the current place of CAD/CAM technology among practicing dentists worldwide is still scarce. What is the attitude of dentists towards this technology? Are they well prepared and adequately educated to deliver such dental service? What about the present nature of dentists’ practice in the provision of CAD/CAM-made dental restorations? All such questions still have no clear answers. The aim of this pilot survey is to investigate attitudes and current practice of dentists in Riyadh, Saudi Arabia regarding chair-side CAD/CAM technology.

## 2. Material and Methods

### 2.1. Study Design

This is a sample survey study. All procedures related to this survey were approved by the research and ethical committee of AlFarabi College of Dentistry and Nursing in Riyadh, Saudi Arabia (IRB: PD01390).

### 2.2. Survey Characteristics

The target population of this survey was practicing dentists in Riyadh city, the capital of Saudi Arabia.

A self-administered questionnaire was designed and piloted among 10 prosthodontists and 10 general dentists to assess clarity of the contents and feasibility of the study. The recorded remarks/comments were utilized to revise the pilot questionnaire. The revised version of the questionnaire was then used in this survey study. The study questionnaire was divided into four sections and presented in English to attract a wider range of participants, as dentists working in Saudi Arabia come from different cultural and educational backgrounds [24]. The first part of the questionnaire was devoted to collect demographic data and information regarding age, gender, nationality, location of dental center in Riyadh, clinical experience, qualification, and specialty of participating dentists. In the second section, four questions related to dentists’ practice and experience with chair-side CAD/CAM system were presented. The third part of the questionnaire comprised nine questions that sought to evaluate attitudes and opinions of the dentists about chair-side CAD/CAM technology. Two questions regarding the need for training on the use of chair-side CAD/CAM concluded the questionnaire questions.

#### 2.2.1. Data Collection

An electronic copy of the questionnaire was prepared using Google forms. A short electronic link was then created and distributed through email and platforms of social media to a convenience sample of dentists working in Riyadh. The aim of the study was presented first and dentists were encouraged to provide their consent and participate by clicking on the attached link to complete the survey items. Confidentiality and anonymity of the collected information were emphasized. The survey was available for completion over more than 6 months, and during this period, at least two reminders to fill in the survey items were sent to non-respondents.

#### 2.2.2. Data Analysis

The SPSS statistical package was used for data analysis (IBM SPSS Statistics for Windows, Version 20.0, Released 2011, IBM Corp, Armonk, New York, NY, USA). Descriptive statistics presented characteristics of participating dentists, and frequency tables were generated to illustrate the response of dentists to survey questions. The Chi-Square statistic was used to assess the association between questionnaire items and dentists’ clinical experience/type of qualification i.e., general dental practitioner versus specialist. A *p*-value < 0.05 was considered significant.

## 3. Results

The questionnaire was sent to 150 dentists. Three of the respondents were working out of Riyadh and omitted, 33 dentists declined completion of the survey items, and 114 questionnaires were completed and included in the final analysis (77.6% response rate). Table 1 presents characteristics of participating dentists. It can be noted that dentists of both genders participated but with a higher proportion of male (57%). The majority of participants (81%) were non-Saudi dentists and participation comprised the different geographical areas of Riyadh city. The clinical experience of participating dentists ranged from 1 to 33 years. General dental practitioners formed a major proportion of the study population (43%) and around 57% of the surveyed dentists were specialists. Almost a third of the participants were prosthodontists or specialists in operative dentistry (32.5%).

Response to practice-related questions indicated that almost two thirds of the respondents operated a chair-side CAD/CAM at some point in the past. Moreover, the chair-side CAD/CAM is available at the current work place of around a quarter of the participants; and a desire to have one was expressed by a significant proportion of participating dentists who do not have a chair-side CAD/CAM at their current practice. In terms of clinical application, a considerable number of the surveyed dentists used the chair-side CAD/CAM in the fabrication of dental crowns (57%). Also, a large number of the participants used the chair-side CAD/CAM in the fabrication of dental inlays and onlays (42% and 37% respectively). However, only a few of the dentists used this machine for the fabrication of fixed partial dentures or implant abutments (16% and 10% respectively). The former results are illustrated in Table 2.

Questions pertinent to attitudes and satisfaction of dentists with chair-side CAD/CAM technology demonstrated a number of interesting findings. Most participants (81%) indicated that the overall quality of chair-side CAD/CAM restorations is at least as good as those fabricated by a lab technician or much better. Around 73% of the dentists evaluated the initial quality of marginal fit, axial contour, proximal contact, and occlusal contact of restorations produced by chair-side CAD/CAM as very good or excellent. Patients’ satisfaction with chair-side CAD/CAM restorations was rated as satisfactory by a considerable proportion of the surveyed dentists (64%). On the dentists’ side, 66% of the participants were satisfied with chair-side CAD/CAM restoration procedure and only a few expressed dissatisfaction (4.4%). More than half of the participating dentists (56%) would likely recommend a chair-side CAD/CAM system to a friend or colleague. The majority of responding dentists considered the chair-side CAD/CAM system as important in terms of time saving, boosting the number of visiting patients and income improvement. The preference for the use of chair-side CAD/CAM system over the conventional restoration system was apparent among three quarters of the surveyed dentists. The abovementioned results are summarized in Table 3.

Table 4 shows the responses of the dentists to two questions about training on the use of chair-side CAD/CAM. The results demonstrate that most of the respondents (87%) feel that it is important for the dentist to carry out training on operating the chair-side CAD/CAM, and a high proportion of them (75%) have the willingness to dedicate the time and effort to learn the chair-side CAD/CAM technology.

The Chi-Square statistic indicated that a significantly larger proportion of general dental practitioners compared to specialist dentists considered that the overall quality of chair-side CAD/CAM restorations is superior to that fabricated by a lab technician (*p* = 0.046). Also, an association was determined between clinical experience of the dentist and rating the importance of training on the use of chair-side CAD/CAM system. It has been found that a significantly greater number of dentists with clinical experience of 10 years or less considered the training as very/extremely important in comparison with dentists who have more than 10 years of clinical experience (*p* = 0.020). No other associations were determined between questionnaire items and dentist’s clinical experience/qualification i.e., general dental practitioner versus specialist.

## 4. Discussion

Despite the major role of CAD/CAM technology in modern dental practice, little information is available about the current practice and attitudes of dentists regarding this innovative technology. In the UK, a survey of dentists was conducted to investigate the status of CAD/CAM technology in UK dental practices [25]. The results indicated no use of any form of digital technology among most of the surveyed dentists. The high cost and absence of perceived merits over traditional methods were considered as barriers for utilization of CAD/CAM technology. Reservations were, also, expressed about the quality of dental restorations produced by chair-side CAD/CAM. Nevertheless, most of the participating dentists indicated that the CAD/CAM technology will have a big place in the future and showed an interest in implementation of this technology in their clinical practice [25]. In Switzerland, a survey of members of the Swiss Dental Association revealed that a chair-side CAD/CAM system was present in 23% of the surveyed practices [26]. A recent study among the USA Navy dental clinics and laboratories demonstrated that, by June 2017, more than a third of the provided indirect restorations (38.1%) were fabricated by a CAD/CAM system [27]. Moreover, analysis of the records illustrated progressive increase in the number of CAD/CAM fabricated restorations over the last 5 years. Due to various advantages, the authors speculated greater implementation of digital dentistry among Navy dentists in the coming future [27]. Few surveys targeted dental students and sought their attitudes towards integration of digital dentistry into dental education and clinical practice, including CAD/CAM technology. The results indicated positive attitudes and a need for greater exposure and integration of CAD/CAM technology into future dental training/education [28,29,30,31].

In Saudi Arabia, in recent years there has been more attention to the value and importance of digital dentistry [32]. Dental professionals in the Saudi job market can feel the growing marketing of CAD/CAM technology on the level of dental companies and continuous professional development courses. As well, many dental centers across Saudi Arabia advertise for their customers the presence of a chair-side CAD/CAM system at their facility as a sign of prestigious oral care services. The current survey can be considered the first to shed some light on attitudes and practices of dental practitioners in Saudi Arabia regarding chair-side CAD/CAM technology.

A recent survey of the commonly used dental materials for indirect restorations among active members of the Saudi Dental Society illustrated that 29.8% of the respondents use the CAD/CAM system in their clinical practice [33]. In our survey, more than a quarter of the surveyed dentists (27.2%) indicated the presence of a chair-side CAD/CAM system at their current work place. On the other side, the desire to have a chair-side CAD/CAM system in the future was quite apparent among the majority of dentists who work in a dental office with no chair-side CAD/CAM system available (Table 2). This is similar to the finding of the British survey as most of the surveyed dentists indicated their interest to incorporate the CAD/CAM technology into their future clinical practice [25]. On the level of clinical practice, the results of this survey show that a substantial number of responding dentists have some experience in operating a chair-side CAD/CAM, particularly for the fabrication of single crowns, inlays and onlays. It seems that the CAD/CAM technology has infiltrated into the workflow of Saudi dental practices with speculations of growing implementation among the wider sector of dental practitioners in the future.

On the level of satisfaction and attitude, the findings of the current survey reflect broad satisfaction and positive attitude amongst participating dentists towards use and outcome of chair-side CAD/CAM in the clinical practice. It can be noted that most participants prefer the chair-side CAD/CAM method over the conventional methods and they appreciate the different merits of the chair-side CAD/CAM system including time saving, income improvement and boosting the number of patients in the clinic. Based on that, it is not surprising that most dentists in this study are ready to recommend a chair-side CAD/CAM system to a friend or colleague. On the contrary to opinions of dentists in the UK survey [25], the majority of dentists in this survey rated positively the overall quality of dental restorations produced by a chair-side CAD/CAM machine. However, it seems that specialist dentists have some concerns about the overall quality of chair-side CAD/CAM restorations, and this merits further investigation.

In the UK survey, a considerable proportion of dentists who use the CAD/CAM technology in their clinical practice regarded their training on this service as not sufficient [25]. In the present survey, most participants, particularly younger dentists, felt that training on the use of a chair-side CAD/CAM machine is important and they have the will to devote the time and effort to learn the chair-side CAD/CAM technology and continue advancing.

A limitation for this study is the relatively small number of participants. This is despite the repeated invitations/reminders to the target population to take part. It can be stated that this survey proved the difficulty of obtaining adequate sample size for an online survey. The shortcomings of such methods for data collection can be recognized. Tools for better outcome should be developed and discussed to overcome the former problem. It has also to be noted that probably the study population comprised dentists who have some experience/are interested in the chair-side CAD/CAM technology and this may have biased the findings.

On the positive side, the collected data came from the various sectors of Riyadh city and presented a random sample of dentists. It comprised both genders, Saudi and non-Saudi dentists, specialists and general dental practitioners, and junior and senior dentists. In addition, conducting the survey in Riyadh adds value to the obtained results. Riyadh is the capital of Saudi Arabia. It has the largest number of dentists and dental centers in Saudi Arabia and hence provides a major bulk of oral/dental health services across the Kingdom [34].

To confirm the findings of this pilot survey, future research should aim and be planned to obtain data on a national level with a representative sample of the dentists in Saudi Arabia.

## 5. Conclusions

The results of this pilot survey reflect broad satisfaction and positive attitude among the surveyed dentists towards use and outcome of chair-side CAD/CAM technology in the clinical dental practice. It seems that the CAD/CAM technology has infiltrated into the workflow of Saudi dental practices with speculations of growing implementation among the wider sector of dental practitioners in the future. 

## Figures and Tables

**Table 1 healthcare-09-00068-t001:** Characteristics of participating dentists (No. = 114).

Age	*Mean (SD)*	*36.2 (7.2)*
Gender	*Male*	*57%*
	*Female*	*43%*
Nationality	*Saudi*	19.3%
	*Non-Saudi*	*80.7%*
Practice location in Riyadh	*North*	29.8%
	*South*	7%
	*Centre*	19.3%
	*East*	36.8%
	*West*	7%
Clinical Experience (years)	*Mean (SD)*	*11.1 (6.8)*
	*≤10 y*	50%
	*>10 y*	50%
Qualification	*DDS/BDS*	43%
	*Postgraduate Diploma*	15.8%
	*MSc*	31.6%
	*PhD*	4.4%
	*Board Certificate*	5.3%
Specialty	*General Practice*	*43%*
	*Prosthodontics*	*19.3%*
	*Operative Dentistry*	*13.2%*
	*Endodontics*	*5.3%*
	*Paediatric Dentistry*	*0.9%*
	*Oral Surgery*	*2.6%*
	*Periodontics*	*4.4%*
	*Oral Medicine*	*0.9%*
	*Orthodontics*	*8.8%*
	*Other*	*1.8%*

**Table 2 healthcare-09-00068-t002:** Response of participating dentists to practice-related questions (No. = 114).

**1. Have you ever operated a chair-side CAD/CAM?** Yes (67.5%) No (30.7%)
**2. Do you have a chair-side CAD/CAM at your current work place?** Yes (27.2%) No (72.8%)
**3. If you do not have a chair-side CAD/CAM at your current work place, do you wish to have one in the future?** Yes (57%) No (15.8%)
**4. Please indicate if you have ever used a chair-side CAD/CAM for the fabrication of any of the following restorations:** Crowns (57%) Bridges (15.8%) Veneers (26.3%) Inlays (42.1%) Onlays (36.8%) Implant abutments (9.6%)

**Table 3 healthcare-09-00068-t003:** Response of participating dentists to attitude-related questions (No. = 114).

**1. How do you evaluate the overall quality of chair-side CAD/CAM restorations in comparison to that fabricated by a lab technician?** Much better than those fabricated by a lab technician (28.1%) Better than those fabricated by a lab technician (18.4%) As good as those fabricated by a lab technician (34.3%) Less than those fabricated by a lab technician (9.6%) Far less than those fabricated by a lab technician (1.8%) I do not know (7.9%)					
**2. How do you evaluate the initial quality of chair-side CAD/CAM restorations in terms of marginal fit, axial contour, proximal contact, and occlusal contact?**					
Excellent (25.4%)	Very good (47.4%)	Good (8.8%)	Fair (0.9%)	Poor (1.8%)	I do not know (15.8%)
**3. Please rate your satisfaction with chair-side CAD/CAM restoration procedure**					
Extremely satisfied (1.8%)	Very satisfied (28.1%)	Satisfied (36%)	Slightly satisfied (1.8%)	Not at all satisfied (4.4%)	I do not know (28.1%)
**4. Based on your clinical experience, please rate patients’ satisfaction with chair-side CAD/CAM restorations**					
Extremely satisfied (7%)	Very satisfied (26.3%)	Satisfied (30.7%)	Slightly satisfied (3.5%)	Not at all satisfied (2.6%)	I do not know (29.8%)
**5. How likely you would recommend a chair-side CAD/CAM system to a friend or colleague?**					
Extremely likely (16.7%)	Very likely (39.5%)	Moderately likely (11.4%)	Slightly likely (7%)	Not at all likely (3.5%)	I do not know (21.9%)
**6. Do you think that a chair-side CAD/CAM system is important in terms of time saving at a dental practice?**					
Yes (75.4%)		No (24.6%)		I do not know (0%)	
**7. Do you think that a chair-side CAD/CAM system is important in terms of boosting the number of patients visiting the dental practice?**					
Yes (76.3%)		No (13.2%)		I do not know (10.5%)	
**8. Do you think that a chair-side CAD/CAM system is important in terms of income improvement?**					
Yes (81.6%)		No (6.1%)		I do not know (12.3%)	
**9. Overall, which restoration system do you most prefer for your practice?**					
Chair-side CAD/CAM system (77.2%)			Conventional system (22.8%)		

**Table 4 healthcare-09-00068-t004:** Response of participating dentists to training-related questions (No. = 114).

**1. How much important do you think training for using a chair-side CAD/CAM?**					
Extremely important (33.3%)	Very important (20.2%)	Important (33.3%)	Slightly important (5.3%)	Not at all important (7.9%)	I do not know (0%)
**2. Are you willing to dedicate the time and effort to learn the chair-side CAD/CAM technology and continue advancing?**					
Yes (75.4%)		No (8.8%)		I do not know (15.8%)	

## Data Availability

The data that support the findings of this study are available on reasonable request from the corresponding author. The data are not publicly available due to privacy or ethical restrictions.

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
