# Peer review of "A Survey of Dentists’ Perception of Chair-Side CAD/CAM Technology"

_healthcare, 2021, doi:10.3390/healthcare9010068_

Round 1

Reviewer 1 Report

Dear author(s),

Thank you for trying to improve our collective knowledge regarding CAD/CAM related attitudes and practice among dental practitioners.

In your sample, 30% of the subjects acknowledged that they have never dealt with chairside CAD/CAM system; therefore it will make sense if you exclude them from the analysis of practice related results as they have no experience with CAD/CAM technology and its products.

In line 24, I'm not sure if "yielding rate" is a clear term. You can consider "response rate".

How did you ensure that no duplication has occurred in your data? You said that you sent multiple invitations and announcements about your survey to the target population.

Please follow the CHERRIS guidelines in reporting your study methods.

Regards,

Author Response

Firstly, we would like to extend our thanks to the editor and reviewers for their valuable and constructive comments.  We confirm that we have gone over all comments, done all required corrections and responded to all queries, as outlined below. Please note that all corrections/revisions in the text have been marked in red font.

Reviewer 1

Comments and Suggestions for Authors

Dear author(s),

Thank you for trying to improve our collective knowledge regarding CAD/CAM related attitudes and practice among dental practitioners.

  • In your sample, 30% of the subjects acknowledged that they have never dealt with chairside CAD/CAM system; therefore it will make sense if you exclude them from the analysis of practice related results as they have no experience with CAD/CAM technology and its products.

Authors’ response:  This is a valid comment. However, in order not to confuse the readers we feel it would be better to keep the results in its current form. Also, questions No 2 and 3 in practice related section ask participants about the presence of CAD/CAM system at their current work place/their desire to have CAD/CAM system at their work place in the future. It is clear that these questions are directed to the whole participants regardless of their practical experience with CAD/ CAM technology.

In line 24, I'm not sure if "yielding rate" is a clear term. You can consider "response rate".

Authors’ response: We agree with this comment. The term has been replaced by: "response rate" and indicated in red font.

How did you ensure that no duplication has occurred in your data? You said that you sent multiple invitations and announcements about your survey to the target population.

Authors’ response: Thank you for this comment. In page 3, lines 112,113 the text has been revised to be: “The survey was available for completion over more than 6 months and during this period at least two reminders to fill in the survey items were sent to non-respondents”.

Please follow the CHERRIS guidelines in reporting your study methods.

Authors’ response: This is very useful. We reviewed CHERRIS guidelines and revised the text accordingly. Please review the methods section in the manuscript. Changes are indicated in red font. 

Reviewer 2 Report

Dear authors, thanks to provide this interesting research. I reported some points to be fixed. Nevertheless, the main problems are the methodology and the english language. The study design is wrong. This is a survey study and not a cross-sectional study. Please check my comments. Then, I suggest a professional editing for english language. There are several repetitions and errors that make the manuscript hard to read. In order to improve the quality of your research, these points should be fixed.

Title should include the specification of the study design. Please include it.

Page 2 line 51 please add company name, city, state in parenthesis.

Lines 67 to 69 please also add:

guided implant surgery, orthognathic surgery, and guided bone regeneration.

Add the following references.

Meloni, S.M., Spano, G., Mattia Ceruso, F., ...Pisano, M., Tallarico, M. Upper jaw implant restoration on six implants with flapless guided template surgery and immediate loadings: 5 years results of prospective case series. ORAL and Implantology, 2019, 12(2), pp. 151–160.

De Riu, G., Virdis, P.I., Meloni, S.M., Lumbau, A., Vaira, L.A. Accuracy of computer-assisted orthognathic surgery. Journal of Cranio-Maxillofacial Surgery, 2018, 46(2), pp. 293–298.

Tallarico, M., Park, C.-J., Lumbau, A.I., ...Koshovari, A., Meloni, S.M. Customized 3D-printed titanium mesh developed to regenerate a complex bone defect in the aesthetic zone: A case report approached with a fully digital workflow. Materials, 2020, 13(17), 3874

Page 2 Lines 71 and 72 please replace constructed with fabricated.

Page 2 lines 74-76 "less need for the dental laboratory, ...reduced consumption of materials, increased productivity" I am not sure this is an advantage. Please clarify/explain or delete.

Page 2 lines 77-78 "higher cost on the patient side" Please explain or delete.

Page 2 line 91 This is not a cross-sectional descriptive study. First of all, cross-sectional study is a "retrospective observational study". The present research is a "sample survey". Please modify thought all the text.

Page 2 lines 93-94 "principles of Helsinki Declaration" must to be adhered to in case of human study (experimental study on human). In case of sample survey, this is not applicable.

Methods. You wrote that "A self‑administered questionnaire was designed and piloted among 10 prosthodontists and 10 general dentists". In the results section you wrote that 150 questionnaires were returned. Please clarify and also report exactly how many questionaries were delivered.

Setting and sample and instrument sections can be merged in only one section "Survey Characteristics".

Author Response

Firstly, we would like to extend our thanks to the editor and reviewers for their valuable and constructive comments.  We confirm that we have gone over all comments, done all required corrections and responded to all queries, as outlined below. Please note that all corrections/revisions in the text have been marked in red font.

Dear authors, thanks to provide this interesting research. I reported some points to be fixed. Nevertheless, the main problems are the methodology and the English language. The study design is wrong. This is a survey study and not a cross-sectional study. Please check my comments. Then, I suggest a professional editing for English language. There are several repetitions and errors that make the manuscript hard to read. In order to improve the quality of your research, these points should be fixed.

 Authors’ response: Thank you so much for your constructive criticism to improve the quality of our manuscript.

Title should include the specification of the study design. Please include it.

Authors’ response: Done. The title has become: A survey of dentists’ perception of chair-side CAD/CAM technology

Page 2 line 51 please add company name, city, state in parenthesis.

Authors’ response: Done and indicated in red font.

Lines 67 to 69 please also add: guided implant surgery, orthognathic surgery, and guided bone regeneration.

Add the following references.

Meloni, S.M., Spano, G., Mattia Ceruso, F., ...Pisano, M., Tallarico, M. Upper jaw implant restoration on six implants with flapless guided template surgery and immediate loadings: 5 years results of prospective case series. ORAL and Implantology, 2019, 12(2), pp. 151–160.

De Riu, G., Virdis, P.I., Meloni, S.M., Lumbau, A., Vaira, L.A. Accuracy of computer-assisted orthognathic surgery. Journal of Cranio-Maxillofacial Surgery, 2018, 46(2), pp. 293–298.

Tallarico, M., Park, C.-J., Lumbau, A.I., ...Koshovari, A., Meloni, S.M. Customized 3D-printed titanium mesh developed to regenerate a complex bone defect in the aesthetic zone: A case report approached with a fully digital workflow. Materials, 2020, 13(17), 3874

Authors’ response: Done and indicated in red font. The references have been inserted in the references section, reference no 13, 14, 15. 

Page 2 Lines 71 and 72 please replace constructed with fabricated.

Authors’ response: Done and indicated in red font.

Page 2 lines 74-76 "less need for the dental laboratory, ...reduced consumption of materials, increased productivity" I am not sure this is an advantage. Please clarify/explain or delete.

Authors’ response: Revised to be: less dependence on the dental technician - - -

Page 2 lines 77-78 "higher cost on the patient side" Please explain or delete.

Authors’ response: The sentence has been deleted.

Page 2 line 91 This is not a cross-sectional descriptive study. First of all, cross-sectional study is a "retrospective observational study". The present research is a "sample survey". Please modify thought all the text.

Authors’ response: Thank you for this comment. In the methods section, the study design has been revised to be: “This is a sample survey study”. Also, revised throughout the text.

Page 2 lines 93-94 "principles of Helsinki Declaration" must to be adhered to in case of human study (experimental study on human). In case of sample survey, this is not applicable.

Authors’ response: Thank you for this comment. It has been deleted.

Methods. You wrote that "A self‑administered questionnaire was designed and piloted among 10 prosthodontists and 10 general dentists". In the results section you wrote that 150 questionnaires were returned. Please clarify and also report exactly how many questionaries were delivered.

Authors’ response: This has been clarified in the results section page 3 lines: 121,122,123 and indicated in red font.

Setting and sample and instrument sections can be merged in only one section "Survey Characteristics".

Authors’ response: Done, materials and methods section, page 2, line 92. 

Reviewer 3 Report

General comments: check the font size of the authors, some are larger than others, some are in bold and others are not; and the corresponding author is missing the affiliation number.

Abstract Section: The abstract is in paragraph format without the titles of the different sections, please remove them and check the coherence of the text without the titles.

Introduction Section:

1) Check the size of the text, for example on page 2, line 69 the words "this innovative" are larger than the rest of the text.
2) Also check the punctuation marks, for example on page 2, line 79: "...for multiple units proshtesis., and...".

Results Section:

1) On page 8, line 166 there is a spelling error "...that he overall quality...". Check the spelling of the whole manuscript just in case.
2) Where you put the result of the p value (page 8, line 167) it is not necessary to put it both ways, it is sufficient to put p=0.046, and the same on line 172.

Author Response

Firstly, we would like to extend our thanks to the editor and reviewers for their valuable and constructive comments.  We confirm that we have gone over all comments, done all required corrections and responded to all queries, as outlined below. Please note that all corrections/revisions in the text have been marked in red font.

General comments: check the font size of the authors, some are larger than others, some are in bold and others are not; and the corresponding author is missing the affiliation number.

Authors’ response: Done.

Abstract Section: The abstract is in paragraph format without the titles of the different sections, please remove them and check the coherence of the text without the titles.

Authors’ response: Done.

Introduction Section:

1) Check the size of the text, for example on page 2, line 69 the words "this innovative" are larger than the rest of the text.

Authors’ response: Done

2) Also check the punctuation marks, for example on page 2, line 79: "...for multiple units proshtesis., and...".

Authors’ response: Done.

Results Section:

1) On page 8, line 166 there is a spelling error "...that he overall quality...". Check the spelling of the whole manuscript just in case.

Authors’ response: Done.

2) Where you put the result of the p value (page 8, line 167) it is not necessary to put it both ways, it is sufficient to put p=0.046, and the same on line 172.

Authors’ response: Revised and indicated in red font, page 8 results section.

Round 2

Reviewer 1 Report

Dear authors,

Thank you for your effort in improving the manuscript. I believe all the points are clarified now.

Regards,